Crizotinib inhibits the metabolism of tramadol by non-competitive suppressing the activities of CYP2D1 and CYP3A2

Gao Nanyong 1 2
Xu Xiaoyu 2
Ye Feng 2
Li Xin-yue 2
Lin Chengqi 3
Shen Xiu-wei wei850916@163.com 4
Qian Jianchang qianjc@wmu.edu.cn 2
1 Yueqing Maternity and Child Health Hospital , Wenzhou , China
2 Wenzhou Medical University , Wenzhou , China
3 Wannan Medical College , Wu hu , China
4 Ruian People’s Hospital , Wenzhou , China
Gould Gwyn
Electronic publication date: 2024 May 30
Publication date: 2024
Volume: 12
Electronic Location ID: e17446
Received 2024 Feb 23; Accepted 2024 May 2
Copyright: ©2024 Gao et al.
Copyright year: 2024
Copyright holder: Gao et al.
License: This is an open access article distributed under the terms of the Creative Commons Attribution License, which permits unrestricted use, distribution, reproduction and adaptation in any medium and for any purpose provided that it is properly attributed. For attribution, the original author(s), title, publication source (PeerJ) and either DOI or URL of the article must be cited.
License URL: https://creativecommons.org/licenses/by/4.0/

Keywords: Tramadol, Crizotinib, Combination, Non-competitive, CYP

Funding: Project of Wenzhou Municipal Science and Technology Bureau Y20220192 The National Natural Science Foundation of China 81973397 The National Key Research and Development Program of China 2020YFC2008301 This work was supported by the Project of Wenzhou Municipal Science and Technology Bureau (Y20220192), the National Natural Science Foundation of China (81973397), and the National Key Research and Development Program of China (2020YFC2008301). The funders had no role in study design, data collection and analysis, decision to publish, or preparation of the manuscript.

==============================
Objectives

To investigate the interaction between tramadol and representative tyrosine kinase inhibitors, and to study the inhibition mode of drug-interaction.

Methods

Liver microsomal catalyzing assay was developed. Sprague-Dawley rats were administrated tramadol with or without selected tyrosine kinase inhibitors. Samples were prepared and ultra-performance liquid chromatography–tandem mass spectrometry (UPLC-MS/MS) was used for analysis. Besides, liver, kidney, and small intestine were collected and morphology was examined by hematoxyline-eosin (H&E) staining. Meanwhile, liver microsomes were prepared and carbon monoxide differential ultraviolet radiation (UV) spectrophotometric quantification was performed.

Results

Among the screened inhibitors, crizotinib takes the highest potency in suppressing the metabolism of tramadol in rat/human liver microsome, following non-competitive inhibitory mechanism. In vivo, when crizotinib was co-administered, the AUC value of tramadol increased compared with the control group. Besides, no obvious pathological changes were observed, including cell morphology, size, arrangement, nuclear morphology with the levels of alanine transaminase (ALT) and aspartate transaminase (AST) increased after multiple administration of crizotinib. Meanwhile, the activities of CYP2D1 and CYP3A2 as well as the total cytochrome P450 abundance were found to be decreased in rat liver of combinational group.

Conclusions

Crizotinib can inhibit the metabolism of tramadol. Therefore, this recipe should be vigilant to prevent adverse reactions.

Introduction

According to statistics from the World Health Organization (WHO), there are approximately 10 million newly diagnosed cancer patients worldwide each year, and the incidence rate is still rapidly increasing, posing a serious threat to people’s lives (Cao et al., 2021; Sung et al., 2021). In this context, tyrosine kinase inhibitors have become the primary first-line therapeutic choice for cancer treatment (Choi et al., 2022; Macia et al., 2013; Mercadante et al., 2022). As a result of cachexia, cancer patients often experience a range of complications, with pain being one of the most common symptoms that reduces a patient’s quality of life. It is estimated that around 70% of patients with advanced cancer suffer from severe pain (Li, Han & Liu, 2022; Motono et al., 2021). Consequently, the combination of tyrosine kinase inhibitors and analgesics is frequently employed as a treatment strategy in clinical settings. However, there are various reports on the effectiveness of combination therapy. Some claim that the combination of both can increase efficacy, while others suggest that they may interact with each other and lead to toxicity and other complications (Shinde et al., 2014; Gadgeel et al., 2007; Varrassi et al., 2022; Yang et al., 2019; Zhao et al., 2017). Therefore, further research is needed on the combination of both.

Tramadol is a synthetic codeine analogue that has obvious analgesic effects and low addiction potential. It has become one of the preferred central analgesics for the treatment of cancer pain, replacing morphine or pethidine in clinical settings (Barbosa et al., 2016; Miotto et al., 2017; Zarghami, Masoum & Shiran, 2012). However, as a µ-opioid receptor agonist and norepinephrine reuptake inhibitor, tramadol can cause common opioid side effects such as gastrointestinal reactions and central nervous system stimulation. It can also easily cause serious side effects such as epilepsy, severe hypotension, hypoglycemia, and adrenal insufficiency (Senthilkumaran et al., 2017; Günther et al., 2018; Ventura, Carvalho & Dinis-Oliveira, 2018).

Tramadol is mainly eliminated through CYP in the liver, and at least 23 metabolites have been identified (Grond & Sablotzki, 2004; Jamali et al., 2017; Perez Jimenez et al., 2016; Wu et al., 2001). Among these, O-desmethyl tramadol can be produced by CYP2D6 pathway metabolism from tramadol. O-desmethyl tramadol has an affinity more than 700 times higher than that of tramadol, and is the substance that primarily activates opioid receptors (Allegaert et al., 2015; De La Gastine et al., 2022; Gillen et al., 2000; Gong et al., 2014). In addition, CYP3A4 and CYP2B6 are also involved in the metabolism of tramadol, and they produce N-desmethyl tramadol, which has no pharmacological activity (Al-Qurain et al., 2021; Grond & Sablotzki, 2004; Grond & Sablotzki, 2004). Therefore, the diverse function of CYP can affect the metabolism of tramadol and lead to differences in efficacy. Since most tyrosine kinase inhibitors are also metabolized through the CYP pathway, there may be drug-drug interactions (DDIs) with tramadol (Abdelhameed et al., 2019; Ding et al., 2013; Jolibois, Schmitt & Royer, 2019). As there are limited reports on this potential interaction, it is essential to investigate this further.

In this study, we screened several representative tyrosine kinase inhibitors to identify potential drug interactions and unveil the underlying mechanism. In addition, considering that Sprague Dawley (SD) rats are common used in the experiment of pharmacokinetics and the homologous gene similarity with human, we used SD rats to carry out the related experiments. The results are expected to provide a theoretical basis for the precise application of tramadol.

Materials and Methods

Chemical and reagents

O-desmethyl tramadol (≥99%, Shanghai Canspec Scientific Instruments Co., Ltd, Shanghai, China); crizotinib, regorafenib and sorafenib (≥98%, Shanghai Canspec Scientific Instruments Co., Ltd, Shanghai, China); 20 other types of tyrosine kinase inhibitors (≥98%, Shanghai Macklin Biochemical Technology Co., Ltd, Shanghai, China); reduced nicotinamide adenine dinucleotide phosphate (NADPH) (≥99%, Shanghai Aladdin Biochemical Technology Co., Ltd., Shanghai, China); phosphate buffered saline (PBS) buffer (Shanghai Beyotime Biology Technology Co., Ltd, Shanghai, China); acetonitrile (ACN), formic acid, and methanol (Sigma-Aldrich, St. Louis, MO, USA); alanine aminotransferase (ALT) and aspartate transaminase (AST) activity assay kits (Nanjing Jiancheng Bioengineering Institute, Jiangsu, China); human liver microsomes (HLM) (Corning Life Sciences Co., Ltd., Jiangsu, China); rat liver microsomes (RLM) were extracted by our team based on the previously reported references (Simpson, 2010).

Equipment and operation condition

Ultra-performance liquid chromatography-tandem mass spectrometry (UPLC-MS/MS) was utilized to measure the concentration of both tramadol and O-desmethyl tramadol. Separation was achieved using a Waters Acquity UPLC BEH C18 column, with dimensions of 2.1 mm × 50 mm and 1.7-µm particle size (Waters Corp., Milford, MA, USA), and quantitation was completed using a Waters XEVO TQD triple quadruple mass spectrometer. The mobile phase was composed of 0.1% formic acid (A) and ACN (B), and the gradient elution was carried out with a flow rate of 0.4 mL/min, following the procedure: 90–10% A (0–1.4 min), 10–90% A (1.4–1.5 min), and 90% A (1.5–2.0 min). Positive mode was used to detect the analytes with the following transitions: m/z 264.2 →58.0, m/z 250.2 →58.2, and m/z 285.0 →154.0 for tramadol, O-desmethyl tramadol, and diazepam (used as an internal standard, IS), respectively. The collision energy was 20 V for tramadol and O-desmethyl tramadol, and 25 V for diazepam. The standard for tramadol and O-desmethyl tramadol were dissolved in a small amount of DMSO (the concentration was 1 mg/mL) then diluted with acetonitrile (the final concentration was 1 ng/mL–1,000 ng/mL). Diazepam was diluted with methanol gradually to 500 ng/mL). All the substance meet the requirements for analysis.

Microsomal incubation assay

To obtain the Michaelis kinetic parameters of tramadol in RLM and HLM, a 200 µL incubation system was established containing 2 µL RLM or HLM (0.2 mg/mL), 186 µL 1xPBS buffer (pH = 7.4), 2 µL tramadol (10–500 µM), and 10 µL NADPH (1 mM). The mixture was pre-incubated for 5 min without NADPH in a water bath shaking at 37 °C. Then, NADPH was added to initiate the reaction, and the mixture was incubated for another 30 min. The reaction was terminated by adding 400 µL of cold ACN and 20 µL of IS (500 ng/mL). After vortexing for 2 min and centrifuging at 16200 × g for 10 min at 4 °C, the supernatant was removed and subjected to UPLC-MS/MS analysis.

For drug-interaction screening, 100 µM of each drug (1.6 µL) was added to the system, and added PBS to the final volume of 200 uL. Tramadol concentration was set at 60 µM according to the Km.

To determine the half-maximal inhibitory concentration (IC50), the concentration of crizotinib was set at 0.01, 0.1, 1, 10, 25, 50, and 100 µM, while the concentration of tramadol was constant at 60 µM in RLM or 100 µM in HLM (according to Km). To determine the underlying mechanism of inhibition, the concentration of crizotinib was set at 0, 4, 8, and 16 µM (RLM) and 0, 5, 10, and 20 µM (HLM) according to the IC50 value, while the concentration of tramadol was set at 15, 30, 60, and 120 µM (RLM) and 25, 50, 100, and 200 µM (HLM) according to the corresponding Km value.

Animal experiment

SD male rats (280 ± 15 g) were purchased from the Shanghai Animal Experimental Center. Before the experiment, the rats were kept in the animal room for two weeks with adequate water and food in order to adapt to the new environment. The room temperature was kept at 20–25 °C and the humidity was kept at 50%–65%. The change period of light and dark conditions was 12 h, which simulates the change of day and night. All animal experiments were approved by the Ethics Committee of Wenzhou Medical University (Approval number: xmsq2022-0621).

Generally speaking, only when at least five or more animals are involved in the experiment can the results be convincing. Thus, twenty SD rats were randomly divided into four groups with five animals per group: group A received a single dose of 20 mg/kg of tramadol; group B received a single dose of 45 mg/kg of crizotinib and 20 mg/kg of tramadol; group C received multiple doses of 0.5% carboxymethylcellulose sodium salt (CMC-Na) for 7 days and a single dose of 20 mg/kg of tramadol; and group D received multiple doses of 45 mg/kg of crizotinib for 7 days and a single dose of 20 mg/kg of tramadol. Prior to the experiment, all rats were fasted for 12 h but allowed free access to water. Once the experiment began, 0.5% CMC-Na and crizotinib were administered to groups C and D for 7 days, respectively, then to groups A and B on the last day of the experiment. After 30 min, all rats were orally administered 20 mg/kg of tramadol. Blood samples were collected from the tail vein at 5 min, 15 min, 30 min, 1 h, 2 h, 3 h, 4 h, 6 h, 8 h, 10 h, 12 h, and 24 h following tramadol administration. During the period, we paid attention to whether the state of rats were normal and no adverse reactions occurred once every hour (if not, we made records and further judged whether it was necessary to be exclude). The samples were centrifuged at 2400 × g for 10 min to obtain serum. Each 50 µL serum was mixed with 150 µL ACN and 20 µL IS (500 ng/mL). After vortexing for 2 min and centrifugation at 16200 × g for 10 min at 4 °C, the supernatant was obtained, then we used UPLC-MS/MS analysis to detect the concentration of tramadol and O-desmethyl tramadol. After the experiment, all the animals were euthanized by 5% isoflurane inhalation anesthesia.

Morphological examination

Nine rats were randomly divided into three groups (n = 3): group E received 0.5% CMC-Na for 7 days; group F received 45 mg/kg crizotinib for 6 days; and group G received 45 mg/kg of crizotinib for 7 days. On the 7th day, all rats were orally administered the substances and then euthanized after 6.5 h (Tmax of crizotinib). The blood, liver, kidney, and small intestine were harvested within 10 min and flash-frozen in liquid nitrogen before being transferred to −80 °C for later use.

Additionally, the tissues were fixed in 10% formalin solution and embedded in paraffin. Hematoxylin-eosin (H&E) staining was performed on paraffin sections, which had been dewaxed, dehydrated, and sealed with neutral gum. Pannoramic MIDI was then used to observe the tissues.

Serum biochemical analysis

Serum was collected after centrifugation of blood sample at 2400 × g for 5 min at 4 °C. The levels of ALT and AST were detected according to the protocol of ALT and AST activity assay kits.

Determine the activities of CYP2D1 and CYP3A2

All liver tissue samples were extracted into liver microsomes by homogenization and centrifugation, following the same operational steps as previously reported (Simpson, 2010). A new incubation system was established, which included 2 µL dextromethorphan or midazolam (the probe substrate of CYP2D1 or CYP3A2), 2 uL RLM (0.2 mg/mL) from different groups, 186 µL 1xPBS buffer, and 10 µL NADPH (1 mM). The concentrations of dextromethorphan and midazolam were set at 25 µM and 10 µM, respectively, according to the Km value obtained from group E. The subsequent steps were identical to those in the ‘Microsomal incubation assay’ section. UPLC-MS/MS was used to detect the concentration of dextrorphan and 1-hydroxymidazolam to determine whether the activities of CYP2D1 and CYP3A2 in the liver were affected by crizotinib.

Carbon monoxide differential UV spectrophotometric quantification

The CO quantitative method was utilized to measure the total quantity of CYP in rat liver microsomes. The microsomes were transferred to a 1.5 mL centrifuge tube and CO gas was introduced for 60 s. Next, sodium dithionite powder was added and mixed thoroughly. After two minutes, the absorbance of the liquid was measured at 450 nm and 490 nm ultraviolet wavelengths using an ultraviolet and visible spectrophotometer to calculate the total amount of CYP.

Statistical analysis

The Michaelis–Menten, IC50, and Lineweaver-Burk plots were generated using GraphPad Prism 6.0 software (GraphPad Software, Boston, MA, USA) with corresponding values for Km, IC50, and Ki (inhibition constant). The mean plasma concentration–time curve for tramadol and O-desmethyl tramadol was drawn using Origin 8.0, and the corresponding pharmacokinetic parameters were obtained using Drug and Statistics (DAS) software (version 3.0). All data were presented as mean ± SD (standard deviation). Statistical differences among the data were calculated using SPSS 24.0 (SPSS, Chicago, IL, USA), with p < 0.05 considered statistically significant. The excessive deviation of data value was considered for exclusion.

Results

Validation of UPLC-MS/MS method for detecting tramadol and O-desmethyl tramadol

Tramadol, O-desmethyl tramadol, and IS were detected using UPLC-MS/MS, and the chromatogram is shown in Fig. S1. The three substances can be well-separated without mutual interference. The ranges of the standard calibration curves for tramadol and O-desmethyl tramadol were 1–1000 ng/mL and 0.1–500 ng/mL, respectively, with correlation coefficients greater than 0.99. To further verify the reliability of the method, we prepared six replicates at low, medium, and high concentrations to assess the accuracy, precision, stability, extraction recovery, and matrix effect of tramadol and O-desmethyl tramadol. The results are shown in Tables S1–S3.

Tyrosine kinase inhibitors, especially crizotinib, can potenially inhibit the metabolism of tramadol

The Michaelis–Menten curve for tramadol in RLM and HLM, along with the corresponding Km value, is shown in Fig. S2. Figure 1A displays the inhibitory effect of 23 tyrosine kinase inhibitors on tramadol metabolism, with crizotinib, sorafenib, and regorafenib exhibiting the highest inhibitory rates of 97.22%, 96.66%, and 83.65%, respectively.

The IC50 curves and Lineweaver-Burk plots of crizotinib on tramadol metabolism are presented in Figs. 1B and 2. The IC50 values were 8.74 ± 0.18 µM and 11.87 ± 0.25 µM in RLM and HLM, respectively. In addition, the Lineweaver-Burk plots indicated that crizotinib may inhibit tramadol metabolism in a non-competitive manner with Ki values of 4.17 µM and 14.40 µM in RLM and HLM, respectively.

Crizotinib suppresses the metabolism of tramadol in SD rats

The mean concentration–time curves of tramadol and O-desmethyl tramadol are shown in Fig. 3, and the corresponding pharmacokinetic parameters are presented in Tables 1 and 2. After a single dose of crizotinib was administered, the values of AUC(0−t) and AUC(0−∞) increased by 45.64% and 55.72%, respectively, compared to the control group for tramadol. However, there were no significant differences observed in the parameters of O-desmethyl tramadol. Upon administration of crizotinib for 7 days, the values of AUC(0−t) and AUC(0−∞) for tramadol increased by 112.01% and 109.01%, respectively, compared to the control group, while CLz/f decreased by 53.47%. Furthermore, the Cmax value for O-desmethyl tramadol decreased by 37.77%. All of the data indicate that crizotinib significantly inhibits the metabolism of tramadol, resulting in changes to its pharmacokinetic parameters.

Figure 1 Determine the interaction between representative tyrocine kinase inhibitors and tramadol.

(A) The inhibitory effect of 23 types of tyrosine kinase inhibitors on the production of O-desmethyl tramadol in RLM compared with the control group. (B) Evaluate the half-maximal inhibitory concentration (IC50) of crizotinib with various concentrations (0.01, 0.1, 1, 10, 25, 50 and 100 µM) on tramadol metabolism in RLM and HLM. Data are presented as the mean ± SD, n = 3.

Figure 2 Primary lineweaver-Burk plot and secondary plot for Ki and αKi in the inhibition of tramadol metabolism by crizotinib with various concentrations in (A) RLM and (B) HLM, respectively.

Data are presented as the mean ± SD, n = 3.

Figure 3 Mean concentration–time curve of tramadol and O-desmethyl tramadol in four groups.

(A) Evaluate the effect of a single dose (45 mg/kg) of crizotinib. (B) Evaluate the effect of multiple doses (45 mg/kg) of crizotinib for 7 days. Data are presented as the mean ± SD, n = 5.

Effect of crizotinib on the tissues morphology and liver function

The results of H&E staining were shown in the Fig. 4A. Compared with the control group (group E), no obvious changes were found in the cell morphology, size, arrangement, nuclear morphology of liver, kidney and small intestine whether crizotinib was administered in a single dose or for 7 consecutive days, which indicated that crizotinib may not cause pathological changes in these tissues.

As shown in Fig. 4B, when crizotinib was administered for a single dose, the level of serum ALT and AST have no significant change. However, when crizotinib was administered for consecutive 7 days, the value of both ALT and AST increased significantly.

Table 1 The main pharmacokinetic parameters of tramadol and O-desmethyl tramadol in group A and group B.

Parameters	Tramadol	O-desmethyl tramadol	
	Group A	Group B	Group A	Group B	
AUC(0−t) (µg/L h)	770.22 ± 168.67	1,121.72 ± 283.26*	934.10 ± 101.30	928.42 ± 144.62	
AUC(0−∞) (µg/L h)	786.25 ± 164.97	1,224.33 ± 344.94*	937.12 ± 101.22	981.18 ± 124.55	
t1/2z (h)	4.22 ± 1.68	6.91 ± 5.73	2.93 ± 0.09	5.68 ± 3.69	
Tmax (h)	0.50 ± 0.00	0.60 ± 0.22	0.80 ± 0.27	0.70 ± 0.27	
Vz/F (L/kg)	167.99 ± 88.22	153.69 ± 96.55	91.31 ± 12.00	169.57 ± 112.02	
CLz/F (L/h/kg)	26.41 ± 5.81	17.89 ± 6.99	21.57 ± 2.60	20.63 ± 2.46	
Cmax (µg/L)	202.00 ± 82.71	244.98 ± 38.33	229.27 ± 56.12	172.26 ± 72.12	
Notes.

Group A: Tramadol single-use group; Group B: Tramadol with crizotinib (single dose).

* P < 0.05, in comparison with the control group.

AUC area under the blood concentration–time curve

t1/2z elimination half time

Tmax peak time

Vz/F apparent volume of distribution

CLz/F blood clearance

Cmax maximum blood concentration

Table 2 The main pharmacokinetic parameters of tramadol and O-desmethyl tramadol in group C and group D.

Parameters	Tramadol	O-desmethyl tramadol	
	Group C	Group D	Group D	Group D	
AUC(0−t) (µg/L h)	777.09 ± 232.62	1,647.49 ± 474.03**	1,172.61 ± 170.75	1,226.57 ± 111.61	
AUC(0−∞) (µg/L h)	790.48 ± 226.30	1,652.18 ± 474.39**	1,175.84 ± 170.44	1,251.86 ± 103.71	
t1/2z (h)	6.69 ± 5.14	2.74 ± 0.56	2.86 ± 0.71	4.06 ± 0.91*	
Tmax (h)	1.60 ± 0.55	0.90 ± 0.65	1.20 ± 0.45	0.90 ± 0.65	
Vz/F (L/kg)	306.53 ± 308.69	51.44 ± 19.09	72.54 ± 25.45	94.72 ± 26.39	
CLz/F (L/h/kg)	27.53 ± 9.92	12.81 ± 3.14**	17.31 ± 2.60	16.07 ± 1.36	
Cmax (µg/L)	245.83 ± 57.58	294.28 ± 85.18	292.40 ± 55.09	181.96 ± 65.78*	
Notes.

Group C: Tramadol single-use group; Group D: Tramadol with crizotinib (multiple dose for 7 days).

*P < 0.05, **P < 0.01, ***P < 0.001 in comparison with the control group.

AUC area under the blood concentration–time curve

t1/2z elimination half time

Tmax peak time

Vz/F apparent volume of distribution

CLz/F blood clearance

Cmax maximum blood concentration

Figure 4 The effect of crizotinib on tissue morphology and liver function.

(A) The results of H&E staining of tissue sections for liver (a), kidney (b) and small intestine (c) in three groups. (B) The value of serum ALT (U/L) and AST (U/L) in three groups. Data are presented as the mean ± SD, n = 3; Vs Control, * P < 0.05, ** P < 0.01.

Crizotinib suppresses the activities of CYP by reducing the abundance of CYP enzymes

The incubation results of dextromethorphan and midazolam are shown in Fig. 5A. When crizotinib was administered, the metabolic rate of dextromethorphan and midazolam decreased to different extents. These results suggest that the activities of CYP2D1 and CYP3A2 in SD rats may be inhibited by crizotinib. Additionally, as seen in Figs. 5B and 5C, the total amount of CYP also decreased upon crizotinib administration.

Figure 5 Crizotinib suppressing the activities of CYP by reducing the abundance of CYP enzymes.

(A) The effect of crizotinib on the function of CYP2D1 (dextromethorphan as probe substrate) and CYP3A2 (midazolam as probe substrate) in the liver of three groups. (B) The chromatogram and (C) the total amount of CYP in rat liver microsomes quantified by CO quantitative method in three groups. Data are presented as the mean ± SD, n = 3; Vs Control, *** P < 0.001.

Discussion

As a commonly used analgesic in clinics, tramadol is often combined with other drugs, which can easily lead to drug-drug interactions (DDIs). Currently, there are reports showing that tramadol tends to interact with some drugs, such as terbinafine, venlafaxine, cimetidine, ketoconazole, and so on (KuKanich, KuKanich & Black, 2017; Saarikoski et al., 2015; Szkutnik-Fiedler et al., 2017). However, there are no reports on the interaction between tyrosine kinase inhibitors and tramadol. Considering that both types of drugs are extensively used in clinical settings, it is meaningful to study the interaction between them and the underlying mechanism.

In this study, we evaluated the effect of 23 types of tyrosine kinase inhibitors on the metabolism of tramadol. We found that crizotinib, sorafenib, and regorafenib had a strong inhibitory effect on tramadol, with an inhibition rate of over 80%. Furthermore, we mainly focused on evaluating the inhibitory effect of crizotinib on tramadol metabolism. The IC50 and Ki values showed that crizotinib can strongly inhibit tramadol metabolism in both RLM and HLM. Besides, the Lineweaver-Burk shows that the straight lines intersect in the negative axis of X-axis (based on the α value), indicated that the underlying mechanism may be non-competitive inhibition.

In order to further study the interaction between crizotinib and tramadol, we carried out animal experiments. We found that whether crizotinib was administered as a single dose or multiple doses for 7 days, there was an increase in both the AUC and Cmax of tramadol, compared to the control group, while the CL decreased to varying degrees. The AUC of O-desmethyl tramadol did not show any significant change, but the Cmax decreased. This effect was more pronounced when crizotinib was administered for 7 days rather than a single dose. It is believed that the inhibition of tramadol metabolism caused by crizotinib resulted in the improvement of bioavailability and the accumulation of tramadol in the body, leading to an increase in its concentration, while the concentration of its metabolite O-desmethyl tramadol still have a similar level. The results were consistent with those obtained in vitro. However, as O-desmethyl tramadol is an active metabolite with a higher affinity to opioid receptors than tramadol, special attention should be paid to the dosage of tramadol when these two drugs are used in combination in order to avoid severe adverse reactions.

At present, crizotinib has been proven to cause liver damage, especially when continuously administered (Duarte et al., 2021; Tsukita et al., 2015; Van Geel et al., 2016). Therefore, we conducted an evaluation of morphological and functional changes in certain tissues after crizotinib administration to further explore why crizotinib inhibits the metabolism of tramadol. We performed H&E staining experiments on liver, kidney, and small intestine to assess morphology. The results indicated that crizotinib did not cause any significant changes in morphology. However, it may have an impact on liver function. After oral administration of crizotinib for 7 days, there was an increase in the levels of ALT and AST, indicating that the liver may have suffered some degree of damage. This is consistent with previous reports (Harada et al., 2021; Tsukita et al., 2015).

Considering that tramadol and crizotinib are primarily metabolized by CYP enzymes (Grond & Sablotzki, 2004; Han et al., 2014; Ng et al., 2022; Shaw et al., 2020). We investigated whether the activity of CYP2D1 and CYP3A2 would be affected by crizotinib. Various studies have shown that crizotinib inhibits CYP3A enzymes (Mao et al., 2013; O’Bryant et al., 2013; Timm & Kolesar, 2013; Xu et al., 2015), which is consistent with our findings. Furthermore, we observed a significant decrease in the metabolic rate of dextromethorphan, a probe substrate of CYP2D1, when crizotinib was administered for seven days. Therefore, we believe that crizotinib has the potential to inhibit the activity of not only CYP3A2 but also part of the activity of CYP2D1. In addition, we conducted CO quantification experiments to evaluate the expression of CYP enzymes in rat liver microsomes. The results indicated that crizotinib could reduce the overall abundance of CYP. This suggests that crizotinib may inhibit CYP activity by reducing the availability of CYP enzymes. As there are no relevant reports yet, we proposed this idea for the first time. Of course, further studies are required to confirm this.

In summary, this study demonstrates a high potential for drug-drug interactions (DDIs) between tramadol and crizotinib. Concurrent use of these drugs can alter the blood exposure of tramadol and impair liver function, leading to serious adverse reactions. Thus, it is advisable to avoid their simultaneous intake. However, this study also has some limitations: We did not evaluate the impact of tramadol on crizotinib metabolism; Additionally, our study was mainly conducted in male SD rats without further discussing the influence of gender differences, which may also can cause drug metabolism difference. Besides, the species differences between rats and humans is also a privacy worthy of attention. Therefore, further clinical studies are warranted to determine the effects of tramadol and crizotinib interaction.

Conclusion

Crizotinib has a potent inhibitory effect on the metabolism of tramadol. Although short-term administration of crizotinib does not lead to toxicity in metabolic organs, the serum levels of ALT and AST increase significantly, accompanied by a reduction in CYP abundance. This collective data could aid in the precise administration of tramadol and crizotinib as personalized medicine.

Supplemental Information

Supplemental Information 1 Raw data

Supplemental Information 2 Arrive 2.0 checklist

We thank Scientific Research Center of Wenzhou Medical University for consultation and instrument availability that supported this work.

Abbreviations

WHO World Health Organization

DDIs drug–drug interactions

RLM rat liver microsome

HLM male human liver microsome

ALK anaplastic lymphoma kinase

PBS Phosphate Buffered Saline

ACN Acetonitrile

ALT Alanine transaminase

AST Aspartate transaminase

UPLC-MS/MS Ultra-performance liquid chromatography–tandem mass spectrometry

NADPH reduced nicotinamide adenine dinucleotide phosphate

CMC-Na carboxymethylcellulose sodium salt

K m Michaelis–Menten constant

IC50 half-maximal inhibitory concentration

K i inhibition constant

H&E hematoxylin-eosin

DAS Drug and statistics

Additional Information and Declarations

Competing Interests

Author Contributions

Animal Ethics

Data Availability

The authors declare there are no competing interests.

Nanyong Gao performed the experiments, authored or reviewed drafts of the article, and approved the final draft.

Xiaoyu Xu performed the experiments, prepared figures and/or tables, and approved the final draft.

Feng Ye performed the experiments, prepared figures and/or tables, and approved the final draft.

Xin-yue Li analyzed the data, prepared figures and/or tables, and approved the final draft.

Chengqi Lin analyzed the data, prepared figures and/or tables, and approved the final draft.

Xiu-wei Shen conceived and designed the experiments, authored or reviewed drafts of the article, and approved the final draft.

Jianchang Qian conceived and designed the experiments, authored or reviewed drafts of the article, and approved the final draft.

The following information was supplied relating to ethical approvals (i.e., approving body and any reference numbers):

The animal study was reviewed and approved by Ethics Committee of Wenzhou Medical University (xmsq2022-0621) on 13-05-2022

The following information was supplied regarding data availability:

The raw measurements are available in the Supplemental Files.

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
