# Peer review of "Crizotinib inhibits the metabolism of tramadol by non-competitive suppressing the activities of CYP2D1 and CYP3A2"

_PeerJ, doi:10.7717/peerj.17446_

## Round 0.1 · original submission · Minor Revisions

As you will see from the comments of the four reviewers, there is a general view that you study makes an important contribution and should be published. All have a few suggestions for improvement which you should work though and amend your paper accordingly (or rebut).

Please make clear in your cover letter what changes you have made in response to each comment.

**Language Note:** The review process has identified that the English language must be improved. PeerJ can provide language editing services - please contact us at [email protected] for pricing (be sure to provide your manuscript number and title). Alternatively, you should make your own arrangements to improve the language quality and provide details in your response letter. – PeerJ Staff

·

Basic reporting

I. Line 135, what was the solvent used to make and tramadol and O-desmethyl tramadol stock solutions for LC-MS measurments?
II. Line 154, was the centrifugation of invitro incubation mixture maintained at room temperature or at cooled to collect supernatant for LC-MS-MS analysis?

Experimental design

I. 2.2. Equipment and operation conditions: for the LC-MS-MS determination of tramadol why diazepam chosen as internal standard over tramadol-d6?
II. 2.3 . In Vivo Experiments: can the accumulated dose of tramadol and O-desmethyltramadol measured to assess toxicity? May need explanation

Validity of the findings

3. Validity of the Findings
I. This manuscript examined Effect of Crizotinib on the Pharmacokinetics of Tramadol and Odesmethyltramadol in rats that would help in assessing clinical applications.
II. LC-MS-MS concentration data for tramadol and O-desmethyltramadol determined to find Pharmacokinetic parameters supports the hypothesis.
III. Conclusions were drawn for the data that was summarized

Reviewer 2 ·

Basic reporting

The manuscript by Gao and colleagues addresses a relevant topic, given cancer incidence and prevalence, the widespread prescription of tramadol and tyrosine kinase inhibitors, as well as the need to clarify drug-drug interactions that may reduce drug effectiveness and increase the likelihood of adverse events, including toxicity.
This manuscript reports a well-reasoned study, combining in vitro and in vivo approaches, with results concerning interrelated biochemical, histopathological, and chromatographic parameters. The results are relevant, correspond to the hypotheses, and have potential real-life application, which represents an added value. The draft is well structured. Adequate literature references and theoretical background are provided.
Despite its current flaws, which I detail below, I believe that this paper fits the scope of PeerJ.

1. The manuscript should be revised for English language, since some verb conjugation and number agreement errors may be found throughout the text. For instance, in lines 32-33: “(…) ultra-performance liquid chromatography-tandem mass spectrometry (UPLC-MS/MS) was used for analysis” should be used instead of “(…) using ultra-performance liquid chromatography-tandem mass spectrometry (UPLC-MS/MS) for analysis”; lines 34-36: “carbon monoxide differential ultraviolet radiation (UV) spectrophotometric quantification was performed” should be used instead of “carbon monoxide differential ultraviolet radiation (UV) spectrophotometric quantification were performed”; line 44: “were found to be decreased” should be used instead of “were found increase”; line 118: “unveil” should be used instead of "unveiled”. These are just a few examples since similar ones can be found throughout the text.
2. Please revise bibliographical citation formatting, since citations appear to not have been automatically or consistently inserted (e.g., lines 102-103: “S et al. 2017; T et al. 2018; Ventura et al. 2018”). Author names appear to be inconsistently abbreviated within the reference list (e.g., line 459: “SM. G, JC. R, EI. H, LK. H, R. V, and A. W. 2007.” versus line 457: “Simpson RJ. 2010.”). Furthermore, expressions such as “What’ more” (line 118) do not seem common or appropriate in a scientific manuscript.
3. Lines 106-107: “The active metabolite of tramadol is O-desmethyl tramadol, which is catalyzed by CYP2D6.” Strictly speaking, O-desmethyl tramadol is not catalyzed by CYP2D6; O-desmethyl tramadol production from tramadol is the one that is catalyzed by CYP2D6.

Experimental design

The research question is well-defined, meaningful, and relevant. There were ethical concerns.

4. Lines 129-130: the procedure for rat liver microsome extraction should be briefly specified, or a bibliographical reference with some details on the procedure should be provided.
5. Section 2.4: How was the total number of animals in the study/number of animals per group defined? Please add some remarks on sample size.

Validity of the findings

All data have been provided, being seemingly robust, controlled and statistically sound. The Conclusions are well stated, linked to the original research question and limited to the results.

6. Please add some remarks on the potential implications of using male animals only in the study design. Could gender-related differences be expected in terms of drug effects?
7. Please suggest one or more mechanisms through which crizotinib can reduce the overall abundance of CYP enzymes, as well as the technique(s) that could be employed to verify that.

Reviewer 3 ·

Basic reporting

no comment

Experimental design

no comment

Validity of the findings

no comment

Additional comments

The study identified the effects of multiple tyrosine kinase inhibitors on tramadol metabolism. The results showed that crizotinib had the most significant inhibitory effect. The authors also investigated the impact of crizotinib on tramadol metabolism after multiple doses. The results suggested that crizotinib reduced the expression of CYP3A and 2D, which was accompanied by liver injury. The article provides a wealth of information, clarifying the potential drug interactions between tyrosine kinase inhibitors and tramadol, and revealing the potential mechanism by which crizotinib affects the metabolism of CYP3A and 2D substrates. I have the following suggestions, which I hope can help the authors revise their paper.
1. In the experiment of drug screening, crizotinib, sorafenib and regorafenib all exhibited the high inhibitory rates on the metabolism of tramadol. Why only crizotinib was selected as the research object in the subsequent experiments? The content of the manuscript will be more attractive if sorafenib and regorafenib are also considered.
2. In the section of “3.3”, the concentration of tramadol was increased significantly when co-administered with crizotinib compared with the control group, which suggest that crizotinib has inhibitory effect on the metabolism of tramadol. However, the concentration of O-desmethyl tramadol seems no downward trend. Please explain the reasons.
3. The metabolic rates of CYP2D1 and CYP3A4 decreased when crizotinib was administered. However, the main question is whether this decrease was due to inhibition of the enzymes' activity or a reduction in their expression levels. Are there any methods that could be used to further verify the underlying cause?
4. Please specify the absolute amount of each component, rather than the volume in the 200 μL incubation system.
5. The formatting of the references is inconsistent. Please revise the references to adhere to a uniform style.

Reviewer 4 ·

Basic reporting

The study clarified the inhibitory effect of commonly used tyrosine kinase receptor inhibitors on tramadol both in vivo and in vitro. In vitro, crizotinib inhibited the metabolism of tramadol in a non-competitive manner. In vivo, the effect of crizotinib on the elimination of tramadol was investigated by single and multiple dosing of crizotinib. The results showed that multiple dosing had a more significant impact, which may be related to the decreased expression of CYP3A4 and CYP2D6, the main metabolic enzymes of tramadol. This study has significant clinical reference value, especially in providing a basis for adjusting the dosage of analgesics for cancer pain patients.

Experimental design

no comment

Validity of the findings

no comment

Additional comments

There are a few issues that need to be addressed to help the author revise the study.
(1)The abstract is overly rigid and lacks coherence, particularly in the results section.
(2)In Section 2.1, the manufacturers and sources of the reagents are not properly labeled. Please revise. For example, Crizotinib (98%, Sigma-Aldrich, St. Louis, MO, USA).
(3)Why was diazepam used as the internal standard?
(4)Why are CYP2D1 and CYP3A2 suddenly mentioned in Section 2.7?
(5)There are issues with the format of Table 1.
(6)The horizontal axis label in Figure 1B is ambiguous and should be the logarithm of concentration. Placing the unit after the label may lead to misinterpretation that the numbers directly represent the concentration.
(7)What does the alpha value in Figure 2 represent, and how can the inhibition type be determined based on the double reciprocal plot results?

---

## Round 0.2 · accepted · Accept

Thank you for attending to the suggestions made in the first round of review. I am content with these changes and am now happy to recommend acceptance.